# The Comprehensive Alcohol Advertising Ban in Lithuania: A Case Study of Social Media

**DOI:** 10.3390/ijerph191912398

**Published:** 2022-09-29

**Authors:** Lukas Galkus, Shannon Lange, Vaida Liutkutė-Gumarov, Laura Miščikienė, Janina Petkevičienė, Jürgen Rehm, Mindaugas Štelemėkas, Alexander Tran, Justina Vaitkevičiūtė

**Affiliations:** 1Health Research Institute, Faculty of Public Health, Lithuanian University of Health Sciences, 47181 Kaunas, Lithuania; 2Institute for Mental Health Policy Research, Centre for Addiction and Mental Health, 33 Ursula Franklin Street, Toronto, ON M5S 2S1, Canada; 3Campbell Family Mental Health Research Institute, Centre for Addiction and Mental Health, 33 Ursula Franklin Street, Toronto, ON M5S 2S1, Canada; 4Department of Psychiatry, University of Toronto, 250 College Street, Toronto, ON M5T 1R8, Canada; 5Department of Preventive Medicine, Faculty of Public Health, Lithuanian University of Health Sciences, 47181 Kaunas, Lithuania; 6Dalla Lana School of Public Health, University of Toronto, 155 College Street, Toronto, ON M5T 1P8, Canada; 7Institute of Clinical Psychology and Psychotherapy, Technische Universität Dresden, Chemnitzer Str. 46, 01187 Dresden, Germany; 8Center for Interdisciplinary Addiction Research (ZIS), Department of Psychiatry and Psychotherapy, University Medical Center Hamburg-Eppendorf (UKE), Martinistraße 52, 20246 Hamburg, Germany; 9Department of International Health Projects, Institute for Leadership and Health Management, I.M. Sechenov First Moscow State Medical University, Trubetskaya Str., 8, b. 2, 119992 Moscow, Russia

**Keywords:** alcohol advertising, social media, Facebook, Instagram, exposure

## Abstract

Alcohol advertising exposure is a risk factor for earlier alcohol initiation and higher alcohol consumption. Furthermore, engagement in digital alcohol marketing, such as liking or sharing an ad on social media, is associated with increased alcohol consumption and binge or hazardous drinking behavior. In light of these challenges, Lithuania has enacted a total prohibition on alcohol advertising, including social media. This study monitored the two most popular social media networks, Facebook and Instagram, to determine compliance with current legislation. In total, 64 Facebook and 51 Instagram profiles were examined. During the 60-day study period, 1442 and 749 posts on the selected Facebook and Instagram profiles, respectively, were published. There were a total of 163 distinct social media alcohol-related posts. Alcohol-related posts accounted for 5.9 percent of total Instagram posts and 8.3 percent of total Facebook posts. Alcohol advertisements accounted for 1.4 percent of all posts (infringement of the Alcohol Control Law). Influencers were responsible for nearly half (45.5 percent) of all observed alcohol-related Instagram posts. The study demonstrates high compliance with Lithuania’s total alcohol advertising ban on social media and emphasizes the importance of adequately monitoring the growing prominence of influencers on social media.

## 1. Introduction

In 2016, Lithuania had the highest per capita alcohol consumption in the European Union (EU) and globally, ranking it highest in the alcohol-attributable mortality and disability-adjusted life years [1]. In response, the Lithuanian government implemented all of the World Health Organization’s (WHO) “best buys” within a short time span—a set of alcohol control policies that are the most effective and cost-effective at reducing alcohol-related harm—which include excise taxation, reduced availability and marketing restrictions [2]. Regarding taxation, there was a sharp increase in alcohol excise tax by 28% on absolute ethyl alcohol (affecting spirits) and 111% on beer and wine between 2016 and 2017. Concerning availability, retail hours for off-premise alcohol sales were shortened from 8 a.m. to 10 p.m. on all days to 10 a.m. to 8 p.m. on Mondays to Saturdays and to 10 a.m. to 3 p.m. on Sundays. Additionally, the minimum legal drinking age was raised from 18 years of age to 20 years of age in 2018 [3].

With respect to marketing restrictions, Lithuania currently stands out as having some of the strictest alcohol advertising laws in the EU [4]. In fact, the WHO deemed Lithuania one of the top three countries in the WHO European region to have the most explicit and probably most effective policies [5]. The Republic of Lithuania Law on Alcohol Control (the Law), enacted on 1 January 2018, provided a comprehensive ban on alcohol advertising, including on all digital media. The official language of the alcohol advertising restriction under Article 29 is provided in Appendix A. In 2020, the WHO reported that they considered the protection of young people with respect to alcohol marketing in Lithuania, Finland and Ireland to be most explicit and probably the most effective in all of the WHO European regions [6]. Yet, it was a long road to reach this point.

Five years after Lithuania declared independence from the Soviet Union on 11 March 1990, the first comprehensive ban on alcohol advertising was adopted with the introduction of the Law. Since its adoption in 1995, the Law has undergone several stricter control and liberalization cycles. In fact, the Law was amended 63 times between 1995 and 2020 [3], including 8 amendments on advertising restrictions alone. A comprehensive timeline and analysis of the Law are published elsewhere [3], whereas the timeline of advertising-related amendments is provided in Table 1.

In their association with public health evidence and ideological positions, alcohol policies were inconsistent since the first time they were introduced. Despite a 1997 decision by the Constitutional Court ruling that alcoholic beverages and tobacco products were “special purpose goods” due to the harm they caused to society, restrictions such as a ban on alcohol advertising could be implemented without being unconstitutional, with the Lithuanian alcohol policy experiencing a lengthy period of liberalization up to 2007. In 2008, for the first time, a comprehensive ban on alcohol advertising was enacted, with a four-year transition period to accommodate the industry. However, after 13 months of intense lobbying by the alcohol industry and its supporters, the ban was revoked 25 days before it went into effect [7]. This illustrated how lobbying and other activities designed to influence policymakers maintained policy environments permissive to alcohol marketing activities [8,9].

Exposure to alcohol advertising is a causal risk factor for earlier alcohol initiation and higher alcohol consumption [10,11]. This potentially incites the industry to gain broad access to younger consumers [12]. An alarming trend in the number of young people exposed to advertisements for alcoholic beverages has been observed over the past several decades as a result of the expansion of alcohol marketing operations on digital platforms. The most recent systematic review concluded that engagement with digital alcohol marketing—such as clicking on an alcohol advertisement (ad), visiting an alcohol-branded website, liking or sharing an ad on social media or downloading alcohol-branded content—is positively associated with increased alcohol consumption and increased binge or hazardous drinking behavior [13]. The alcohol industry stated that social media marketing can reach more consumers than broadcast media, with a 600% return on investments [13].

To this day, to our knowledge, no studies have been conducted investigating the frequency with which alcohol is advertised on social media or determining whether the advertising ban in Lithuania is being adhered to or not. This study aims to determine the compliance of the total alcohol advertising ban on social media by monitoring the two most popular social media networks—Facebook and Instagram. Our hypothesis is that current alcohol advertisement ban legislation and efficient enforcement mechanisms are in place to deter producers and retailers from marketing their goods.

## 2. Materials and Methods

Our study was quantitative and descriptive in design. The choice of such a study method was primarily determined by the intent to observe and describe the current situation of alcohol advertisement on social media without any external interference.

The most popular social media networks in Lithuania [14], Facebook and Instagram, were monitored for alcohol-related posts for 30 consecutive days and 30 days retrospectively. Figure 1 shows the flowchart of the study design process.

### 2.1. Pilot Phase and Preparation for Data Collection

In January 2021, two data collectors were introduced to the study protocol and trained to collect data. Before data collection, pilot tasks were given to both data collectors to evaluate inter-rater reliability in assessing potential alcohol ads [15]. The data collectors were given 24 h to independently review the 14-day social media history on two Facebook pages and two Instagram accounts and identify potential alcohol ads. In total, 64 posts were published, of which 5 met the criteria to be captured (criteria described under the section ‘Monitoring of Social Media’). The data collectors were instructed not to communicate with each other about the task.

Out of 64 posts reviewed, the first data collector correctly identified (based on the given criteria) all five social media posts as potential alcohol ads and misreported one social media post, and the second data collector classified all social media posts correctly. The evaluation resulted in an overall agreement of 98.4% of all social media posts. Using Cohen’s kappa statistic, inter-rater reliability analysis was performed to determine inter-coder reliability. The result (κ = 0.901) indicated an almost perfect degree of inter-coder agreement.

### 2.2. Study Phase: Monitoring of Social Media

In total, we monitored five categories of social media accounts: 10 of the most profitable grocery retailers [16], 3 of the most popular specialized liquor chain stores, 10 shopping malls (i.e., all malls operating in Lithuania), 23 alcohol producers and 18 of the most followed influencers [17]. Data collectors reviewed 64 Facebook and 51 Instagram profiles in total (Table 2).

The data collectors looked for alcohol-related posts that met at least one of the inclusion criteria: (1) the post mentioned alcohol and alcohol look-alike brand(s); (2) posted an image or video portraying alcohol or alcohol look-alike beverage bottle(s), glass(es), can(s) or had alcohol-related objects (e.g., the cork of a champagne bottle); and (3) the post mentioned a category of alcoholic beverage (e.g., wine, beer, etc.).

Selected profiles on Facebook and Instagram were examined for 30 days, starting from 27 January to 26 February 2021. In addition, on the first day of prospective data collection in the study, researchers examined all posts posted in the preceding 30 days and took screenshots of all potential ads that met at least one of the inclusion criteria. Thereby, a total of 60 days of posts was examined. The data collectors reviewed posts published on social media profiles, including Instagram stories, every day at 4 pm to ensure that posts lasting 24 h were captured. Given that content published on weekends and holidays was examined on the first following business day, Instagram stories were not captured from approximately 4 pm on Friday to 4 pm on Sunday due to only being accessible for 24 h, nor were they collected in a retrospective data collection.

Data collection was performed in Microsoft Excel, reporting on the date of the post, day of the week the post appeared, brand name, type of alcoholic beverage, type of zero-alcohol beverage, social media site, hyperlink to the post and a screenshot of the post. The principal investigator was available via telephone and email for specific queries from the data collectors during the entire study.

For all captured posts, the number of ‘likes’, ‘comments’ and ‘shares’ (on Facebook posts) was recorded. These indicators were not recorded for Instagram, as ‘shares’ are not available on Instagram and the ‘likes’ count is hidden in Lithuania.

### 2.3. Study Phase: Reviewing and Grouping of Social Media Posts

Four investigators independently grouped alcohol-related posts into one of the following three categories: (1) alcoholic beverage ad, (2) zero-alcohol beverage ad (in this study defined as beverages that looked alike to alcoholic beverages in terms of packaging and display, and likely to be advertised because of similarity to alcoholic counterparts) and (3) “grey zone”, such as product placement in a social setting, for example. The categorization took place when at least three out of the four investigators designated a particular post to the same category. All cases where less than three investigators agreed on the categorization were discussed until a consensus was reached.

The research team established the abovementioned categories to reflect local alcohol industry marketing tactics. Social media posts in the first category contained alcoholic beverage brand names (written or pictorial) and were considered a potential infringement of the current regulation. The second category included ads for zero-alcohol beverages, such as alcohol-free beer, alcohol-free wine, etc. Alcohol producers started to exploit legal loopholes by labelling and packaging both categories of drinks similarly after the introduction of the TV ad restrictions in 2007, which became more evident after the total ban in 2018. Posts were categorized as being in the “grey zone” when they showcased alcohol in a positive social context or displayed alcoholic drinks, but no brand could be identified.

After the primary screening, investigators removed nine ads that were duplicates and 13 that did not fit under any of three categories due to a lack of strong association with alcohol (e.g., an ad portraying an empty wine glass among other objects with no alcohol-related text).

Differences in the distribution of posts between Facebook and Instagram were assessed using the Chi-square (*χ2*) test. *p*-values of less than 0.05 were considered as statistically significant.

## 3. Results

Within the 60-day study period, 1442 and 749 posts were published on the selected Facebook and Instagram profiles, respectively. In total, 163 distinct social media posts constituted our final sample of alcohol-related posts (Table 3). Similar to the total number of screened posts, the number of alcohol-related posts was higher on Facebook. Alcohol-related posts composed 5.9% (*n* = 44) of total posts on Instagram and 8.3% (*n* = 119) on Facebook (*p* = 0.03). In addition, 121 alcohol-related stories were posted on Instagram.

Altogether, out of all published posts, 1.4% (*n* = 30) of posts was categorized as alcohol ads (infringement of the Law), 1.2% (*n* = 26) posts as zero-alcohol beverage ads and 4.8% (*n* = 107) as “grey zone” ads.

Compared to Instagram (5.9%), a greater share of alcohol-related posts was published on Facebooks accounts (8.3%) (*p* = 0.014). The share of alcohol-related posts published during workdays was higher on Facebook (92.4%) compared to Instagram (81.8%) (*p* < 0.001) (Table 3).

By only analyzing alcoholic beverage ads (Category 1), it could be observed that most posts on Facebook and Instagram were posted by alcohol producers, comprising 85.0% and 90.0%, respectively (Table 4). Similarly, if we were to search for accounts with the greatest share of alcohol-related content, on both Facebook and Instagram, the accounts of alcohol producers had more than one in three posts that were related to alcohol (38.6% and 36.5%, respectively).

Almost half (48.0%) of all observed zero-alcohol beverage ads were posted by grocery retailers. Specialized liquor stores and shopping malls posted zero-alcohol beverage ads in very similar proportions, 28.0% and 24.0%, respectively. On Instagram, the highest number (58.9%) of “grey zone” ads was posted by influencers, and, on the contrary, by alcohol producers on Facebook (58.1%) (Table 4).

Although influencers were the most active group on social media in general in terms of the number of posts published, the frequency of alcohol-related posts was relatively low on both platforms (1.6% on Facebook and 3.9% on Instagram). However, even with a low share of published alcohol-related posts among all monitored profiles, influencers posted almost half (45.5%) of all observed alcohol-related posts on Instagram.

Within the study period, alcohol-related posts on Facebook were ‘liked’ 14,126 times, commented on 9619 times and shared 4691 times. Facebook users interacted the most with ads for zero-alcohol beverages. The engagement with zero-alcohol beverage ads was the highest in terms of all three parameters, with a median ‘likes’ count of 143 (range 0–1263), a median ‘comments’ count of 180 (range 0–1800) and a median ‘shares’ count of 123 (range 0–588). Meanwhile, posts of alcoholic beverages (category 1) were far less popular in terms of ‘likes’, ‘comments’ and ‘shares’ (Table 5).

Influencers generated the most alcohol-related stories, with more than 8 out of 10 alcohol-related stories (84.3%) posted on their profiles (Table 6). There were only 11 alcoholic beverage ads posted as stories, with the majority (81.8%) posted by alcohol producers. Similar to posts on Instagram, the highest number (93.1%) of “grey zone” ads was posted by influencers. 

## 4. Discussion

This was the first study examining compliance with a comprehensive alcohol advertising ban on social media in Lithuania. Overall, 1.4% of all published posts on Instagram and Facebook that we reviewed was categorized as alcohol ads, violating legal restrictions, and most of them were posted by alcohol producers. If zero alcohol beverage and ‘grey zone’ ads are included, there were 7.4% of all the posts published having some link or appearing to be an alcohol ad. User engagement with advertisements for alcoholic beverages was lower than their engagement with advertisements for zero-alcohol beverages or with posts displaying drinking behavior. Even though the number of alcohol-related posts published by influencers was very low (considering the fact that influencers generate a large number of posts regularly), overall, those posts accounted for almost half of all alcohol-related posts on Instagram among the analyzed profiles.

Current legislation in Lithuania forms a strong restrictive marketing environment not commonly found in other countries and in the area of research. The already published studies analyzing the role of social media mostly focused on examining patterns of alcohol producers, retailers and influencers using social media to promote alcohol, as well as studies exploring the content of alcohol ads, while researching compliance and the enforcement of a total ban was likely under-researched purely due to the fact that there are only a few countries (Kazakhstan, Kyrgyzstan, Norway and the Russian Federation) with such regulations. As there may be growing support for tighter regulations [18,19,20], it is likely that marketing restrictions could also become a more important research topic.

One of the most frequent arguments used by the alcohol industry and its allies in 2011, when lobbying to revoke a comprehensive alcohol advertising ban, was that there were no existing instruments capable of regulating content on social media [7]. At the time, one of the industry’s key groups, the Lithuanian Free Market Institute, stated that “young people are best reached by advertising on the internet”, and that “it is uncertain whether and how the advertising ban will affect alcohol consumption in this age group” [21]. Such arguments were no longer utilized in June 2021, when a group of parliamentarians filed a change to the Law to allow alcohol producers and retailers to advertise alcohol on their social media accounts if the posts were not sponsored [22]. This was the most recent and unsuccessful attempt to loosen the restrictions on alcohol advertising, which was met with strong opposition from national and European public health organizations [23].

Our findings contradicted the industry’s claims of the impracticability of enforcing a total advertising ban, as only 1.4% of the 2191 posts was classified as alcoholic beverage ads. Furthermore, according to the Drug, Tobacco and Alcohol Control Department, which oversees the implementation of the ban, alcohol advertising accounted for only 1.7% of all identified infringements of the Law in 2018 [24]. The share of infringements remained relatively low in the following years, at 4.2% in 2019, 4.9% in 2020 and 2.1% in 2021 [24]. Even with a small team of five civil employees working on this issue in the Drug, Tobacco and Alcohol Control Department, it appeared that enforcing a total ban is possible. The Drug, Tobacco and Alcohol Control Department focuses on retailer education and tries to prevent the infringement of new regulations by providing between 3500 and 4500 alcohol advertising consultations to businesses and issuing approximately 90 warning letters each year [25]. Compliance specialists also monitor social media networks. In 2019, 240 unique profiles were examined for compliance, and 300 in 2020 [25]. High compliance may be related to a few other reasons. The controlling authority is allowed to block noncompliant websites or social media profiles and levy civil penalties ranging from EUR 2896 to EUR 21,721 (EUR 6019 on average) [24]. Moreover, social media networks have their internal advertising policies that are developed in accordance with local laws. For example, on Facebook, ads are automatically reviewed by the system for violations of advertising policies [26] and are deleted if they are in violation of current legislations.

According to our study’s findings, a similar number of alcohol ads and zero-alcohol beverage ads were posted on Facebook and Instagram (30 and 26, respectively). In 2008, when new restrictions on alcohol advertising were introduced, businesses started promoting zero-alcohol beverages that looked almost identical to alcoholic drinks. Zero-alcohol beers and kvass, which were sold in the same containers and with the same labels as ordinary beer, were promoted by alcohol manufacturers. Later, the industry cited this as an example of inadequate regulation to argue for the liberalization of the legal framework. Producers of the spirits openly expressed their concern that beer producers continued to promote actual beer brands by promoting alcohol-free beer, and that this was how they exploited regulatory loopholes [27]. Moreover, the market for zero-alcohol beverages nearly doubled in 2018 compared to the previous year, with zero-alcohol cider sales increasing by 160% and beer and wine sales increasing by 90% [28]. Due to the market for zero-alcohol beverages currently expanding and with sales of all types of alcohols shrinking in Lithuania since 2014, it is difficult to predict whether advertisements for zero-alcohol beverages can be utilized (as demonstrated before) to take advantage of regulatory gaps in the future and to revert the current trend.

The Supreme Administrative Court of Lithuania (the Court) has now twice, in a highly contentious manner, interpreted civil penalties imposed by the Department for potentially utilizing zero-alcohol beverages to promote alcoholic beverages. A well-known spirit producer started the production of cranberry soft drinks that were labelled as “Čepkelių zero-alcohol nonbitter”, but that was visually almost identical to “Čepkelių cranberry bitter, 36% volume”, and the Supreme Administrative Court of Lithuania ruled that the producer had violated the law by marketing a beverage that was visually identical to an alcoholic spirit drink and, thus, indirectly promoting it [29]. Meanwhile, when the beer producer advertised “Utenos” zero-alcohol beer cans, the appearance of which was identical to the alcoholic beer cans, the Supreme Administrative Court of Lithuania reversed the decision of the regional administrative court and concluded that the Law did not prohibit the advertising of zero-alcohol beverages, nor were there any requirements for the trademark of zero-alcohol beverages, and the element of alcohol advertising in the dispute was purely hypothetical [30]. Such contradictory court rulings fail to provide legal precedent for evaluating the use of zero-alcohol beverages to promote alcoholic beverages. The lack of consistency in legal practice caused this to become an important area for future studies.

One of the main objectives for further research on this subject is to elucidate how sales of alcoholic and zero-alcohol beverages (such as alcohol-free beer) changed when advertising was banned. Such studies should aim to clarify whether zero-alcohol beverage advertising can be both indirectly advertising alcoholic beverages and advertising encouraging the consumption of alcohol-free versions of those beverages, which should in theory result in a sharp increase in zero-alcohol beverage sales. These findings would facilitate the development of more specific recommendations for decision makers regarding the need to amend (or maintain the status quo) current legislation to avoid advertising ban loopholes. Although such an analysis would be of benefit to this field, it was beyond the scope of our study.

According to the data of this study, influencers posted half (45.5%) of all alcohol-related posts on Instagram and 84.3% of all Instagram stories. However, only two (10% of posts and 9.1% of stories on Instagram) of these cases were categorized as alcohol ads, since they clearly displayed the brand. The remaining were categorized as being in the “grey zone” category, which refers to photos, videos or live broadcasts that did not contain any direct or indirect allusions to a specific alcohol brand, but rather focused on a positive social drinking environment. A study conducted in the Netherlands [31] showed similar results on influencers actively posting about alcohol. This is disturbing because influencers are proven to be popular among children and minors [32], and influencer marketing elicits less opposition than traditional means of advertising [33]. Fully aware of the growing use of celebrities and influencers by the alcohol industry [25], the Lithuanian Drug, Tobacco and Alcohol Control Department published guidelines for influencers to help them determine whether the information they plan to post on social media could be considered an alcohol ad [34]. Similar practices of the industry financially engaging influencers to promote alcohol brands and make drinking more desirable were recorded elsewhere [35]. As evidence of social media’s influence on alcohol intake and attitudes continues to mount [36], it is necessary to find new ways to regulate activities of this kind in order to safeguard public health.

This study had some limitations. First, the data were collected over a short period of time. Consequently, it was impossible to compare the data and identify patterns. In addition, the data were collected during the COVID-19 pandemic with strict lockdown measures in Lithuania. In the presence of dining, indoor gathering and night club restrictions, it could be assumed that alcohol-related posts were published less frequently compared to a restrictions-free setting. On the other hand, studies showed an increase in alcohol consumption during the pandemic lockdown in Lithuania [37]. It could be argued that the lockdown could have been used by retailers and producers to promote alcoholic beverages as part of leisure activities whilst at home. Additionally, the categorization of alcohol-related posts was based on local realities and the method of allocating posts to different categories was not validated. In addition, not reviewing social media pages on weekends may have precluded the capture of posts that were quickly removed after they were published. The same was applicable to Instagram stories, which disappeared from the feed and profile after 24 h. Furthermore, even though we were able to record ads for alcoholic beverages, due to the characteristics of Facebook and Instagram, we were unable to determine what was the actual engagement of posts published (i.e., their ‘reach’ or the amount of ‘views’). To account for this limitation, we evaluated the popularity of captured Facebook posts in terms of ‘likes’, ‘comments’ and ‘shares’. The popularity of captured posts, however, was likely to continue to increase with time with potentially more people encountering the post and, therefore, could not be fully recorded.

Despite these constraints, our study provided valuable insight into the extent of alcohol ads on social media in the setting of a comprehensive alcohol advertising ban. As long as the Law remains one of the most frequently amended legal acts in Lithuania, more in-depth independent research is of paramount importance to help proban groups to prevent the further liberalization of the Law. Future studies should seek to better understand the content of published alcohol-related posts and what the characteristics are that increase user engagement on social media. In addition, future research should attempt to evaluate the effects of the total ban on the advertising of alcohol in Lithuania.

## 5. Conclusions

Data from our study were encouraging in terms of demonstrating compliance of alcohol producers, retailers and influencers with Lithuania’s total alcohol advertising ban on social media. Moreover, our study highlighted the need for regulatory responses towards zero-alcohol beverages that were nearly identical to alcoholic beverages of the same brand. Additionally, there is a need to properly continuously monitor the expanding prominence of influencers’ social media accounts, as they likely contribute significantly to posts that potentially promote positive associations with drinking. Future studies should consider exploring these nuances and potential tactics of alcoholic beverage advertisers to identify potential gaps within the efficient enforcement of a total advertising ban.

## Figures and Tables

**Figure 1 ijerph-19-12398-f001:**
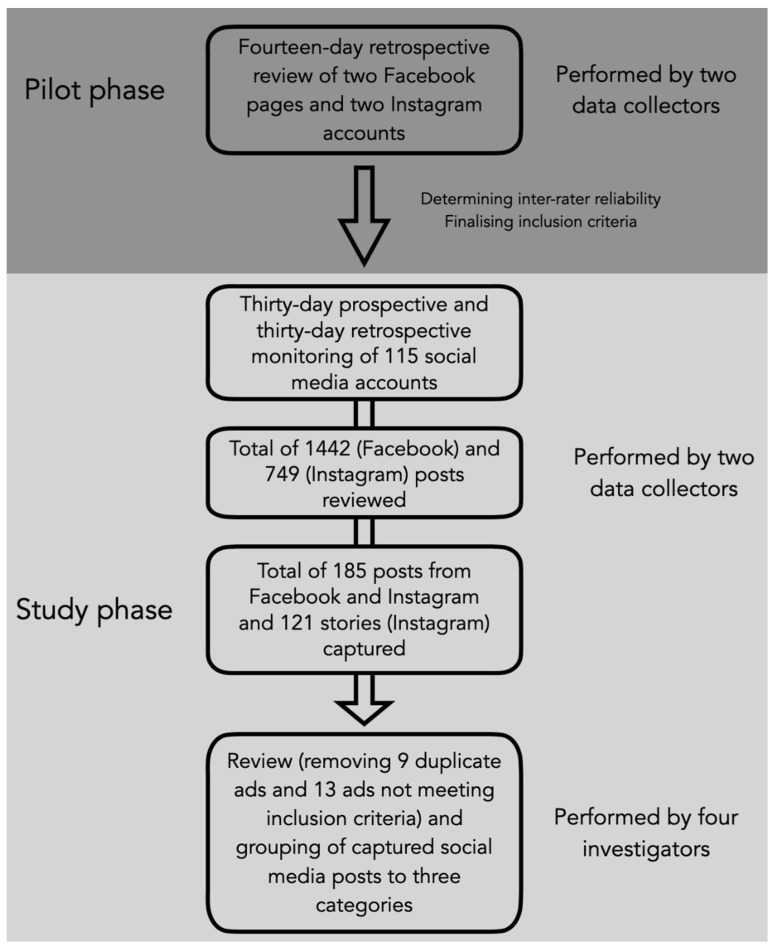
The flowchart of the study design process.

**Table 1 ijerph-19-12398-t001:** Developments of the regulation of alcohol advertising in Lithuania.

Adopted	Came Into Effect	Measures Introduced	“+” in Favor of Public Health “-” against Public Health
18 April 1995	26 May 1995	The first nearly comprehensive ban on alcohol advertising was adopted within the introduction of the Law in 1995. The ban clearly pinpointed the areas for advertisements: locally broadcasted radio and television programs, print media, specialized advertising brochures as well as indirect advertising. However, these measures were not strictly enforced.	+
2 July 1997	16 July 1997	The Law was further amended to redefine areas of the ban of advertisements. A comprehensive ban was narrowed down. The placement of advertisements was prohibited on national television and radio programs from 3 p.m. to 10 p.m., on weekends and school holidays from 8 a.m. to 10 p.m., on other television and radio programs from 3 p.m. to 8 p.m., and on weekends and school holidays from 8 a.m. to 8 p.m. (excluding beer and wine with alcohol strengths of less than 15%). Outdoor alcohol advertising was permitted.Amendments introduced a ban related to the content of ads: prohibition/restriction advertising targeting children under 18, and false or misleading information about alcoholic beverages. The amendment also detailed what is not permitted in the ad itself.	-
20 June 2002	28 June 2002	The Law allowed the advertising of alcohol below a volume strength of 22% on television and radio programs. The placement of advertisements on television and radio programs was prohibited from 3 p.m. to 10:30 p.m., and on weekends and school holidays from 8 a.m. to 10:30 p.m. Outdoor advertising was banned for all alcoholic beverages except for wine, beer and cider.	+/-
1 July 2003	16 July 2003	The display of alcoholic beverages was permitted in nearly all event venues and petrol stations. Trademark logos and names were excluded from the advertisement definition (i.e., permitted to be displayed).	-
21 June 2007	1 January 2008	Alcohol advertising banned on TV and radio during daytime hours (6 a.m. to 11 p.m.).	+
18 April 2008	Did not come into effect. Law was revoked on 6 December 2011	Amendment to the Law for a comprehensive ban on alcohol advertising from 2012 was passed but not implemented.	
17 May 2016	1 November 2016	Amendment prohibited organizing games, actions, competitions or lotteries to promote the purchase or use of alcohol, and alcohol was not to be used as a prize, gift or bonus. The promotion of alcohol price reductions was prohibited.	+
1 June 2017	1 January 2018	A comprehensive ban on alcohol advertising of all forms, including social media, was introduced.	+

**Table 2 ijerph-19-12398-t002:** Summary of monitored Facebook and Instagram profiles by category.

	Profiles		
Category	Facebook	Instagram *	Total
Grocery retailers	10	7	17
Specialized liquor stores	3	3	6
Shopping malls	10	7	17
Alcohol producers	23	16	39
Social media influencers	18	18	36
Total	64	51	115

* Not all selected retailers, producers or influencers had both Facebook and Instagram profiles. Therefore, the number of monitored Instagram profiles is lower.

**Table 3 ijerph-19-12398-t003:** Summary of key results by weekday (*n* = 163).

Weekday	Total *n* (%)	Total Alcohol-Related *n* (%)	Category 1 (Alcoholic Beverage Ad) *n* (%)	Category 2 (Zero-Alcohol Beverage Ad) *n* (%)	Category 3 (Grey Zone) *n* (%)
Facebook posts					
Monday	185 (12.8)	20 (16.8)	1 (5.0)	10 (40.0)	9 (12.2)
Tuesday	258 (17.9)	15 (12.6)	1 (5.0)	4 (16.0)	10 (13.5)
Wednesday	236 (16.4)	28 (23.5)	5 (25.0)	6 (24.0)	17 (23.0)
Thursday	249 (17.3)	28 (23.5)	6 (30.0)	2 (8.0)	20 (27.0)
Friday	267 (18.5)	19 (16.0)	5 (25.0)	1 (4.0)	13 (17.6)
Saturday	137 (9.5)	3 (2.5)	1 (5.0)	1 (4.0)	1 (1.4)
Sunday	110 (7.6)	6 (5.0)	1 (5.0)	1 (4.0)	4 (5.4)
Total	1 442 (100)	119 (100)	20 (100)	25 (100)	74 (100)
Instagram posts					
Monday	107 (14.3)	5 (11.4)	0 (0.0)	0 (0.0)	5 (15.2)
Tuesday	99 (13.2)	11 (25.0)	4 (40.0)	0 (0.0)	7 (21.2)
Wednesday	131 (17.5)	6 (13.6)	0 (0.0)	0 (0.0)	6 (18.2)
Thursday	130 (17.4)	9 (20.5)	2 (20.0)	1 (100.0)	6 (18.2)
Friday	118 (15.8)	5 (11.4)	1 (10.0)	0 (0.0)	4 (12.1)
Saturday	77 (10.3)	4 (9.1)	2 (20.0)	0 (0.0)	2 (6.1)
Sunday	87 (11.6)	4 (9.1)	1 (10.0)	0 (0.0)	3 (9.1)
Total	749 (100)	44 (100)	10 (100)	1 (100)	33 (100)
Facebook and Instagram posts					
Total	2 191 (100)	163 (7.4)	30 (1.4)	26 (1.2)	107 (4.8)

**Table 4 ijerph-19-12398-t004:** Summary of key results by social media profile category (*n* = 163).

Category of Social Media Profile	Number of Posts *n* (%)	Number of Alcohol-Related Posts *n* (%)	The Proportion of Alcohol-Related Posts from Total Published Posts (% of Total)	Category		
				Category 1 (Alcoholic Beverage Ad) Posts *n* (%)	Category 2 (Zero-Alcohol Beverage Ad) Posts *n* (%)	Category 3 (Grey Zone) *p n* (%)
			Facebook posts			
Grocery retailers	464 (32.2)	24 (20.2)	5.2	0	12 (48.0)	12 (16.2)
Specialized liquor stores	44 (3.1)	13 (10.9)	29.5	2 (10.0)	7 (28.0)	4 (5.4)
Shopping malls	282 (19.5)	14 (11.9)	5.0	0	6 (24.0)	8 (10.8)
Alcohol producers	158 (11.0)	60 (51.3)	38.6	17 (85.0)	0	43 (58.1)
Influencers	494 (34.2)	8 (6.7)	1.6	1 (5.0)	0	7 (9.5)
Total	1442 (100)	119 (100)	8.3	20 (100)	25 (100)	74 (100)
			Instagram posts			
Grocery retailers	123 (16.4)	0	0	0	0	0
Specialized liquor stores	12 (1.6)	0	0	0	0	0
Shopping malls	44 (5.9)	1 (2.3)	2.3	0	0	1 (2.9)
Alcohol producers	63 (8.4)	23 (52.3)	36.5	9 (90.0)	1 (100.0)	13 (38.2)
Influencers	507 (67.7)	20 (45.5)	3.9	1 (10.0)	0	19 (58.9)
Total	749 (100)	44 (100)	6	10 (100)	1 (100)	33 (100)

**Table 5 ijerph-19-12398-t005:** Number of ‘likes, ‘comments’ and ‘shares’ per alcohol-related post category on Facebook within 60 days.

Category of Alcohol-Related Post	‘Likes’ (n)					‘Comments’ (n)					‘Shares’ (n)				
	Min	Max	Percentiles			Min	Max	Percentiles			Min	Max	Percentiles		
			25	50	75			25	50	75			25	50	75
Category 1 (Alcoholic beverage ad)	0	702	6	29	63	0	31	0	1	6	0	11	0	1	3
Category 2 (Zero-alcohol beverage ad)	0	1263	0	143	346	0	1800	0	180	356	0	588	0	123	285
Category 3 (Grey zone)	0	1713	8	25	61	0	368	0	0	1	0	250	0	0	3
Total	0	1713	5	25	100	0	1800	0	0	6	0	588	0	1	7

**Table 6 ijerph-19-12398-t006:** Summary of alcohol-related stories by Instagram profile category (*n* = 121).

	Total Alcohol-Related Stories Count *n* (%)	Category 1 (Alcoholic Beverage Ad) Count *n* (%)	Category 2 (Zero-Alcohol Beverage Ad) Count *n* (%)	Category 3 (Grey Zone) Count *n* (%)
Grocery retailers	1 (0.8)	0	1 (14.3)	0
Specialized liquor stores	3 (2.5)	1 (9.1)	1 (14.3)	1 (1.0)
Shopping malls	1 (0.8)	0	0	1 (1.0)
Alcohol producers	14 (11.6)	9 (81.8)	0	5 (4.9)
Influencers	102 (84.3)	1 (9.1)	5 (71.4)	96 (93.1)
Total	121 (100)	11 (100)	7 (100)	103 (100)

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
