# Peer review of "The Comprehensive Alcohol Advertising Ban in Lithuania: A Case Study of Social Media"

_ijerph, 2022, doi:10.3390/ijerph191912398_

Round 1

Reviewer 1 Report

Thank you for the opportunity to review the manuscript „The comprehensive alcohol advertising ban in Lithuania: a case study of social media“. This study monitored the two most popular social media networks, Facebook and Instagram, to determine compliance with current legislation.

However, I have some concerns: 

What type of research was carried out? (quantitative or qualitative?)

What was the design of this study?

According to the recruitment process of this study, the authors should add a flow diagram of this study.

What were the research questions?

The statistical methods used for data analysis appear insufficient to generalise results into the conclusions.

Generally, both aim and hypotheses of this study were unclear. The design of the study was also unclear. Additionally, only univariate analysis was carried out. The conclusions were similar to the general considerations. The discussion section lacks a clear framework for discussion.

In addition, authors forgot to interpret the results in the context of data published by authors from other countries. It is necessary to increase the number of references so that the issue of this study can be fully disclosed.

I would suggest that these comments might be taken into account and that the manuscript should be changed. However, the idea authors set is interesting.

Kind Regards

Reviewer 2 Report

The current manuscript entitled "The comprehensive alcohol advertising ban in Lithuania: a case 2 study of social media" is well written and presented. Also highlighting the importance of alcohol advertising ban in Lithuania.

I have major concern about data collection & time period

This data collection has been done during the peak of COVID-19 and it cannot be generalized to normal condition.

What is the inclusion and exclusion criteria of adds

Why choose only Facebook and Instagram? What about other social media and TV/magazine adds

Why Zero volume alco-holic beverage category they used for add bans? It contains alcohol? What is the role of regulatory authorities if it contains alcohol and marked as zero volume alcoholic beverages.

This study should be compare with data of normal condition (away from COVID-19 period) for the better results.

Statistical significant results of chi square test?

Conclusion need to more detailed about ban on advertisement of alcohol 

Reviewer 3 Report

This study monitored the two most popular social media networks, Facebook and Instagram, to determine compliance with current legislation in Lithuania. In total, 64 Facebook and 51 Instagram profiles were examined. During the 60-day study period, 1,442 and 749 posts on the selected Facebook and Instagram profiles, respectively, were published. There were 163 distinct social media alcohol-related posts. Alcohol advertisements accounted for 1.4 percent of all posts (infringement of the Alcohol Control Law). Finally, the study concluded that high compliance with Lithuania’s total alcohol advertising ban on social media and emphasizes the significance of adequately monitoring the growing prominence of influencers on social media.

Overall, the manuscript was well written. However, a few concerns/comments needed to be explained/modified. 

  1. Line 167-169 is whether those drinks have alcohol or not
  2. In result, It would be nice if the authors could prepare a flow chart of the study design for the common reader of your manuscript.
  3. Line 232-233 I think the percentage was very less, what do the authors think about it
  4. Line 257 why there were few people
  5. Line 270-271 where was the reference cited?
  6. Please check the journal style, it should be in short form.

Round 2

Reviewer 1 Report

Dear Authors,

In my opinion, you have addressed all my questions. The topic you are writing is new and relevant. I recommend this manuscript for acceptance.

Best Regards

Reviewer 2 Report

Authors improved manuscript as per suggestions. 

I have no more other suggestions